# A Serine Protease Inhibitor, Camostat Mesilate, Suppresses Urinary Plasmin Activity and Alleviates Hypertension and Podocyte Injury in Dahl Salt-Sensitive Rats

**DOI:** 10.3390/ijms242115743

**Published:** 2023-10-30

**Authors:** Yasunobu Iwata, Qinyuan Deng, Yutaka Kakizoe, Terumasa Nakagawa, Yoshikazu Miyasato, Miyuki Nakagawa, Kayo Nishiguchi, Yu Nagayoshi, Yuki Narita, Yuichiro Izumi, Takashige Kuwabara, Masataka Adachi, Masashi Mukoyama

**Affiliations:** 1Department of Nephrology, Kumamoto University Graduate School of Medical Sciences, Kumamoto 860-8556, Japan; 2Comprehensive Clinical Education, Training and Development Center, Kumamoto University Hospital, Kumamoto 860-8556, Japan; 3Department of Pharmacy, Kumamoto University Hospital, Kumamoto 860-8556, Japan

**Keywords:** camostat mesilate, podocyte, epithelial sodium channel, hypertension, plasmin

## Abstract

In proteinuric renal diseases, the serine protease (SP) plasmin activates the epithelial sodium channel (ENaC) by cleaving its γ subunit. We previously demonstrated that a high-salt (HS) diet provoked hypertension and proteinuria in Dahl salt-sensitive (DS) rats, accompanied by γENaC activation, which were attenuated by camostat mesilate (CM), an SP inhibitor. However, the effects of CM on plasmin activity in DS rats remain unclear. In this study, we investigated the effects of CM on plasmin activity, ENaC activation, and podocyte injury in DS rats. The DS rats were divided into the control diet, HS diet (8.0% NaCl), and HS+CM diet (0.1% CM) groups. After weekly blood pressure measurement and 24-h urine collection, the rats were sacrificed at 5 weeks. The HS group exhibited hypertension, massive proteinuria, increased urinary plasmin, and γENaC activation; CM treatment suppressed these changes. CM prevented plasmin(ogen) attachment to podocytes and mitigated podocyte injury by reducing the number of apoptotic glomerular cells, inhibiting protease-activated receptor-1 activation, and suppressing inflammatory and fibrotic cytokine expression. Our findings highlight the detrimental role of urinary plasmin in the pathogenesis of salt-sensitive hypertension and glomerular injury. Targeting plasmin with SP inhibitors, such as CM, may be a promising therapeutic approach for these conditions.

## 1. Introduction

In the renal collecting duct, the activation of the epithelial sodium channel (ENaC) occurs through the proteolytic cleavage of its γ subunit by extracellular serine proteases (SPs), which are primarily regulated by aldosterone in physiological conditions [1,2]. However, in proteinuric kidney disease, plasminogen—which leaks into the tubular lumen through damaged glomerular barriers—is converted to the serine protease plasmin via the effect of urokinase-type plasminogen activator (uPA). Subsequently, plasmin activates γENaC in the collecting duct, which leads to aldosterone-independent sodium retention—a process known as the “overflow theory” in nephrotic syndrome [3,4,5]. Additionally, plasminogen can be transformed to plasmin by uPA on the cell surface of podocytes, causing hemodynamics-independent podocyte injury [6,7]. Therefore, plasmin is considered to be involved in the pathogenesis of hypertension and glomerular injury in proteinuric kidney disease.

Camostat mesilate (CM), a synthetic serine protease inhibitor developed in Japan, has been used clinically for several decades to treat conditions such as chronic pancreatitis and reflux esophagitis after surgery [8]. We have previously demonstrated the blood pressure-lowering and renoprotective effects of CM in rodent models of hypertension and chronic kidney disease (CKD) [9,10,11,12,13]. In our past study, we observed that plasmin is activated in the kidney of aldosterone and salt-treated rats, and CM mitigated hypertension and renal damage in association with the inhibition of plasmin activity [12]. Similarly, in high-salt (HS)-loaded Dahl salt-sensitive (DS) rats, which leads to the development of severe hypertension and proteinuria, together with plasma aldosterone-independent inappropriate γENaC activation, CM ameliorated these effects [9,14]. Recently, we reported that treatment with a synthetic plasmin inhibitor, YO-2, reduced γENaC activation and podocyte damage in DS rats, further underscoring the potential of targeting plasmin as a treatment way for salt-sensitive hypertension [15]. However, it remains unclear whether the beneficial effects of CM previously observed in DS rats are directly associated with the inhibition of plasmin activity. Therefore, in this study, we administered CM to HS-loaded DS rats to evaluate its effects on plasmin activity, ENaC activation, and glomerular injury.

## 2. Results

### 2.1. The Effects of CM on Body Weight, Organ Weight, Systolic Blood Pressure, Urinary Protein, and Blood Parameters

After 5 weeks, the HS group resulted in a reduction in body weight against the control group; however, there were no significant differences between the HS and HS+CM groups (Appendix A). The systolic blood pressure (SBP) consistently and significantly increased from weeks 2 to 5 in the HS group; in contrast, CM treatment mitigated this increase (Figure 1A).

The HS group exhibited substantial proteinuria, which was significantly attenuated by CM (Figure 1B). Both PRA and PAC were markedly suppressed by the HS diet, while CM had no effect, which is consistent with our previous findings (Figure 1C) [13]. Cardiac and renal hypertrophies were provoked in the HS group, whereas CM administration tended to reduce cardiac hypertrophy and significantly suppressed renal hypertrophy (Table 1). The renal dysfunction observed in the HS group was alleviated by CM treatment. The CM treatment significantly reduced serum sodium and chloride levels against the HS group (Table 1).

### 2.2. The Effects of CM on Urinary Plasmin

The urinary plasmin activity and protein abundance, as evaluated by double-layer fluorescent zymography and immunoblotting, were increased by HS diet (Figure 1E,F). The increased urinary plasmin activity was confirmed using the plasmin-specific chromogenic substrate chromozym PL (Figure 1D). The CM treatment reduced the abundance of plasminogen/plasmin and urinary plasmin activity (Figure 1D–F).

### 2.3. The Effects of CM on Renal ENaC Expression

In the HS group, the mRNA expressions of βENaC and γENaC, except for αENaC, were upregulated, and CM treatment did not suppress these inductions (Figure 2A). Immunoblotting revealed that the protein abundances of βENaC and both the full-length band and cleaved band of γENaC were induced by the HS diet, while αENaC remained unaffected (Figure 2B,C), as we have reported [14]. Interestingly, CM not only reduced the cleavage of γENaC, but also decreased the abundance of the full-length protein, as well as βENaC (Figure 2B–D). The immunofluorescent staining confirmed that γENaC and aquaporin 2 (AQP2), a representative marker of the collecting duct, co-localize in the collecting duct cells (Appendix A). The HS loading induced the apical trafficking of γENaC, which was not apparently affected by CM treatment (Figure 2B and Appendix A). Since the downregulation of epidermal growth factor (EGF) in DS rats due to HS loading has been reported to increase the protein abundances of βENaC and γENaC [16], we evaluated the expressions of EGF and its receptor (EGFR). The mRNA expression of EGF decreased, whereas that of EGFR increased in the HS group; CM administration significantly restored EGF expression (Figure 2A). These findings suggest that the restoration of EGF signaling may be associated with the reduced protein abundances of βENaC and γENaC due to CM administration.

### 2.4. The Effects of CM on Glomerular Sclerosis, Podocyte Injury, and Glomerular Apoptosis

Next, we evaluated the effects of CM on HS-induced damages in glomeruli of DS rats. Glomerular sclerosis and plasminogen/plasmin co-localized with synaptopodin, which is a major podocyte marker that was decreased in the HS group, were observed at 5 weeks (Figure 3 and Appendix A).

These changes were markedly alleviated by CM (Figure 3 and Appendix A), suggesting that the presence of attached plasmin(ogen) is associated with podocyte injury in DS rats, which is consistent with our previous report [15]. In the glomeruli, the number of apoptotic cells determined with TUNEL staining was increased, while the number of WT-1-positive podocytes was decreased by HS loading (Figure 4A,B). Since WT-1 is a marker of healthy podocytes and it is reduced in injured podocytes, most of the TUNEL-positive podocytes may have already lost WT-1 staining. Therefore, it is difficult to observe the double staining of both markers in the same single podocyte. The protein levels of apoptotic signaling molecules, both the Bax/Bcl-2 ratio and cleaved caspase-3, were increased by HS loading (Figure 4D,E); CM treatment suppressed these changes (Figure 4A–E).

### 2.5. The Effects of CM on Protease-Activated Receptors (PARs) and Kidney Injury Markers

Since plasmin adversely affects kidney injury progression through PAR activation, and PAR-1 and PAR-4 have been implicated in podocyte damage in rats [17], we investigated their expressions in the renal tissue of DS rats. The protein abundance of PAR-1, but not PAR-4, was increased by HS loading (Figure 5A,B). Extracellular signal-regulated kinase (ERK), a downstream molecule of PAR-1, was more phosphorylated by HS loading (Figure 5C,D). CM treatment significantly suppressed these changes (Figure 5A–D). The renal injury markers were increased by HS loading, and CM treatment attenuated these effects (Figure 5E).

## 3. Discussion

Proteinuric renal diseases are characterized by the activation of urinary serine proteases, particularly plasmin, which activates ENaC in the distal renal tubule through the cleavage of its γ subunit. This activation causes sodium retention and hypertension [4]. In addition, plasmin leakage induces podocyte injury through various mechanisms, including oxidative stress and apoptosis [6,7,18]. Our previous studies showed that DS rats loaded with an HS diet developed hypertension and proteinuria, accompanied by proteolytic activation of γENaC, and that treatment with CM attenuated hypertension and proteinuria [9,14]. Moreover, we recently showed that treatment with a synthetic plasmin inhibitor, YO-2, mitigates γENaC activation, hypertension, podocyte injury, and glomerular sclerosis in DS rats [15]. However, whether the beneficial effects of CM are related to plasmin inhibition in DS rats remains to be elucidated. In the current study, CM treatment suppressed urinary plasmin activity, inhibited γENaC activation, and alleviated salt-sensitive hypertension. Furthermore, CM treatment attenuated podocyte injury by preventing the co-localization of plasmin and podocytes, and by reducing the number of glomerular apoptotic cells. These findings suggest that plasmin may be a crucial target for the therapeutic effects of CM in DS rats.

In the DS rats, both βENaC and γENaC were aberrantly upregulated and γENaC was proteolytically activated by the HS diet, even in the presence of suppressed plasma aldosterone levels, as observed in past studies [14,16]. In the current study, CM treatment not only inhibited the cleavage of γENaC, but also suppressed the protein abundance of βENaC and γENaC. Our recent study using the plasmin inhibitor YO-2 demonstrated that it suppressed the cleavage of γENaC without affecting its protein abundance [15]. Therefore, CM may have additional effects on the regulation of ENaC. A previous study revealed that the downregulation of EGF by the HS diet contributes to increased abundances of βENaC and γENaC in the kidneys of DS rats [16]. Consistent with this, in this study, EGF expression was suppressed by the HS diet, and CM treatment restored its expression. This suggests that the restoration of EGF expression due to CM treatment may contribute to the suppression of βENaC and γENaC abundance. We previously reported that CM treatment partially inhibited γENaC activation without affecting its expression in aldosterone-infused rats [10]. Although the reason for this discrepancy is unclear, it is possible that the expression and role of EGF differ between DS rats with low plasma aldosterone and aldosterone-infused rats. Furthermore, it is unknown whether the effect of CM treatment on EGF is due to its serine protease inhibition or other properties, such as scavenging oxidative stress [11]. Further investigations are necessary to elucidate the detailed mechanisms underlying these findings.

The role of plasmin in ENaC activation appears to be critical in DS rats. In addition, there are several reports to suggest the importance of plasmin in the development of sodium retention and hypertension in human disease [19,20,21,22,23,24,25,26,27,28]. However, its significance in experimental nephrotic mice remains controversial. In knockout mice of plasminogen and uPA, sodium retention and the activation of γENaC were observed in the nephrotic state, similar to wild-type nephrotic mice, despite the absence of active plasmin in the urine [29,30]. This suggests that other serine proteases may play roles in ENaC activation when plasmin is absent. Conversely, another group reported that neutralizing antibodies against uPA inhibited urinary plasmin activation and γENaC cleavage in the kidneys of nephrotic mice [31,32]. Additionally, although CM treatment did not inhibit ENaC activation in nephrotic mice, another broad-spectrum serine protease inhibitor, aprotinin, mitigated the activation of ENaC and sodium reabsorption in the renal tubule [33]. These findings suggest that the proteases responsible for ENaC activation differ among species. Therefore, it is important to conduct more studies to make these findings applicable to hypertension treatment in humans.

Plasminogen leakage through damaged glomeruli is also activated to plasmin by the action of uPA, leading to podocyte injury independent of its effects on ENaCs, thereby accelerating the development of proteinuria [6,7]. Our previous study demonstrated that CM suppresses the activity of plasmin in the renal tissue, podocyte injury, as well as interstitial fibrosis in aldosterone- and salt-loaded uninephrectomized rats [12]. Additionally, CM mitigated podocyte apoptosis, via a partly blood pressure-independent manner, in high-salt diet-fed metabolic syndrome rats [13]. In this study, we observed the co-localization of plasmin(ogen) with podocytes, as determined by the podocyte marker synaptopodin, and found that CM inhibited their co-localization and alleviated podocyte loss. A plasmin inhibitor, YO-2, exerted similar effects in our recent study, further supporting the notion that plasmin is the primary target of CM in podocyte injury [15]. Moreover, we have previously shown that CM attenuates the progression of renal dysfunction in another CKD model [11]. Therefore, serine protease inhibitors targeting plasmin, including CM, may become novel therapeutic ways in the treatment of CKD.

Our study has some limitations. Firstly, we could not elucidate the mechanism by which CM treatment restores EGF expression. Although some serine proteases, such as prostasin, are known to modulate EGFR signaling through proteolytic modifications [34,35], there have been no reports demonstrating the involvement of serine proteases in EGF expression. Considering that plasmin is involved in the maturation of inflammatory cytokines and other soluble factors [36,37], it is conceivable that certain serine proteases may be involved in the maturation and regulation of EGF. Further investigations are required to clarify the relationship between serine proteases and EGF expression. Secondly, although plasmin appears to be a primary therapeutic target of CM in this experiment, it cannot be ruled out that other proteases besides plasmin may be involved in the context of ENaC activation and podocyte injury. In relation to the activation of ENaC, it has been reported that numerous serine proteases such as prostasin, kallikrein, and uPA are excreted in the urine along with proteinuria, and may activate ENaC [19,38,39,40]. As CM inhibits these proteases, it is possible that other target serine proteases aside from plasmin may be involved. When it comes to podocyte injury, while a recent study demonstrated that plasmin is also involved in the podocyte injury in doxorubicin-induced nephropathy [18], a coagulation serine protease thrombin has been reported to injure podocytes in nephrotic rats and diabetic nephropathy rats by activating PAR-1 and PAR-4 [18,41]. Although we did not detect thrombin activity in urine or kidney tissue using specific substrates or DLF zymography (Boc-Val-Pro-Arg-MCA, No. 3093-v, Peptide Institute, data not shown), CM could suppress thrombin activity and alleviate thrombin-associated podocyte injury in DS rats. Furthermore, in diabetic nephropathy, podocyte injury has been reported to involve other serine proteases, such as factor X (FX) and complement factor B [42,43]. Nevertheless, given the similarity between the experimental results using the plasmin-specific inhibitor YO-2 reported previously and the results with CM in this study [15], plasmin is likely the main target protease of CM in Dahl rats. Further studies using different serine protease substrates, inhibitors, and knockout mice are required to elucidate the roles of other serine proteases in salt-sensitive hypertension. As a third point of concern, it remains unclear whether the results of this study can be extrapolated to humans. In previous studies, CM has been reported to ameliorate urinary abnormalities in conditions such as glomerulonephritis and diabetic nephropathy [44,45,46,47]. However, these clinical trials have been conducted on a small scale, and the impact of CM on blood pressure has not been observed. The dosage of CM used in this animal experiment is approximately ten times higher per body weight compared to the amount typically used in clinical settings (In rats, the dosage is approximately 80 to 100 mg/kg, while in humans, it ranges from 5 to 10 mg/kg in the clinical setting.). To achieve similar results in humans, it may be necessary to administer higher doses of CM. In any case, to ascertain whether CM can be an effective treatment for salt-sensitive hypertension in humans, it is essential to conduct prospective large-scale clinical trials, including the evaluation of dosages.

In conclusion, the results of this study suggest that plasmin is involved in the ENaC activation and the exacerbation of podocyte injury in salt-sensitive hypertension, and protease inhibitors targeting plasmin could potentially become a novel treatment for salt-sensitive hypertension with proteinuria (summarized in Figure 6).

## 4. Materials and Methods

### 4.1. Animal Experiments

All of the animal experiments were conducted following the approved guidelines at Kumamoto University (No. A2021-096) and the ARRIVE (Animal Research: Reporting of In Vivo Experiments) guidelines. Three-week-old male DS rats were purchased from the Charles River Laboratories Japan (Kanagawa, Japan) and housed with free access to water and food under controlled conditions of temperature (22 ± 1 °C), humidity (55 ± 2%), and 12-h light–dark cycles. The CM was supplied by Ono Pharmaceutical Co., Ltd. (Osaka, Japan).

At 4 weeks of age, the rats were divided into the following three groups: (1) the control group (0.3% NaCl diet; n = 4); (2) the HS group (8.0% NaCl diet; n = 6); and (3) the HS+CM group (8.0% NaCl + 0.1% CM diet, n = 6). The CM was administered orally by mixing it into the diet at a concentration of 0.1%. Each rat was placed in a metabolic cage for 24-h urine collection once per week. The systolic blood pressure (SBP) was measured weekly using the tail-cuff method (MK-2000; Muromachi Kikai Co., Ltd., Osaka, Japan). After 5 weeks, the rats were anesthetized as previously described [15], and blood samples were obtained via the inferior vena cava. Then, both kidneys of all rats were removed and sliced into approximately 3 mm thick sections for further analyses. Serum creatinine, electrolytes, total protein, plasma renin activity (PRA), plasma aldosterone concentration (PAC), and urinary protein were measured in a commercial laboratory (SRL, Tokyo, Japan).

### 4.2. Pathological Examination

The kidneys were sectioned, fixed in 4% paraformaldehyde, followed by embedding in paraffin. Sections of approximately 3 μm thickness were stained with periodic acid–Schiff (PAS) for the microscopic evaluation of morphological changes. The glomerulosclerosis score was calculated based on the evaluation of approximately 50 glomeruli [14]. Five or more randomly selected sections were evaluated in a blinded manner.

### 4.3. Urinary Serine Protease Activities

To evaluate urinary serine protease activity, a urine sample (corresponding to the 2000th part of the 24-h urine volume) from each rat was mixed with Laemmli buffer in a non-reduced condition and applied to sodium dodecyl sulfate–polyacrylamide gel electrophoresis (SDS–PAGE). DLF zymography with a serine protease substrate, QAR-MCA (No 3135-v, Peptide Institute Inc., Osaka, Japan), was performed to assess urinary plasmin activity as per previously described methods [12,15,48]. The MCA released by plasmin activity was visualized using an ultraviolet transilluminator at a wavelength of 365 nm. The same urine sample was used for evaluation of urinary albumin in silver staining (silver staining MS kit; Fujifilm, Osaka, Japan).

### 4.4. Chromogenic Assay for Urinary Plasmin Activity

We measured urinary plasmin activity using a plasmin-specific substrate, chromozyme PL (Tosyl-Gly-Pro-Lys-4-nitroanilide acetate, Sigma-Aldrich, Burlington, MA, USA), which was cleaved by plasmin, resulting in the generation of a residual peptide and 4-nitroanilin. The urine samples normalized to creatinine concentrations were incubated in a reaction buffer containing chromozyme PL. The released 4-nitroaniline was then measured with a spectrophotometer at a wavelength of 465 nm.

### 4.5. Western Blotting

For membrane protein extraction (for ENaC, PAR-1, and PAR-4), we homogenized a single piece of kidney tissue using a Polytron (PT 10-35 GT) in an isolation solution buffer consisting of 250 mM sucrose, 10 mM triethanolamine, 1 g/mL leupeptin, and 0.1 mg/mL phenylmethylsulfonyl fluoride (all purchased from Sigma-Aldrich). The homogenate was subjected to differential centrifugation at 1000× *g* for 10 min and 17,000× *g* for 30 min, followed by centrifugation at 200,000× *g* for 1 h at 4 °C to obtain membrane proteins, as previously described [10]. The obtained pellets were dissolved in RIPA buffer (Fujifilm, Osaka, Japan) mixed with a protease inhibitor cocktail (Sigma-Aldrich).

Another piece of kidney tissue was homogenized in a tissue protein extraction reagent (T-PER, Thermo Scientific, Rockford, IL, USA) mixed with a protease inhibitor cocktail to obtain whole-kidney protein (for Bax, Bcl-2, cleaved caspase-3, ERK, and actin). Another piece of kidney tissue was also processed in anti-phosphatase buffer (25 mM HEPES, 10 mM Na_4_P_2_O_7_·10H_2_O, 100 mM NaF, 5 mM EDTA, 2 mM Na_3_VO_4_, 1% Triton X-100, and protease inhibitor cocktail) to maintain the phosphorylation of proteins (for phospho-ERK1/2). The protein concentration was measured using the Pierce BCA protein assay kit (Thermo Scientific).

Thirty micrograms of kidney samples were subjected to SDS–PAGE using a 12.5% constant gel or a 4–12.5% gradient gel. Then, the proteins were transferred onto nitrocellulose membranes (GE Healthcare Life Sciences, Chicago, IL, USA) and blocked with 5% skim milk powder (Fujifilm, Osaka, Japan) or 1% BSA for phosphorylated-ERK1/2 for 30 min at room temperature. We incubated membranes with primary antibody at 4 °C overnight, then with secondary antibodies at room temperature for 1 h. Thereafter, protein blots were incubated in ECL prime Western blotting detection reagents (GE Healthcare) for 1 min, and images were visualized using an Amersham Imager 680 (GE Healthcare). The primary antibodies used were anti-plasminogen (ab154560, Abcam, Cambridge, UK) and anti-ENaC (anti-α: SPC-403, anti-β: SPC-404, anti-γ: SPC-405, StressMarq Biosciences Inc., Victoria, BC, Canada). Antibodies against Bax (sc-493), Bcl-2 (sc-492), PAR-1 (sc-13503), PAR-4 (sc-1666), and actin (sc-1616) were obtained from Santa Cruz Biotechnology (Santa Cruz, CA, USA). The anti-cleaved caspase-3 (D175), anti-ERK1/2 (4695S), and anti-phospho-ERK1/2 (p-ERK1/2) (4370S) antibodies were purchased from Cell Signaling Technology (Danvers, MA, USA).

### 4.6. Immunofluorescence

The kidney tissue was embedded in Tissue-Tek OCT compound (Sakura Finetek Japan Co., Tokyo, Japan) and frozen in liquid nitrogen, then carefully sliced into thin sections measuring 3 μm in thickness. To ensure proper handling, the sections were air dried using silica gel for 30 min. Next, the sections were fixed in 4% paraformaldehyde for 20 min and washed with PBS for 5 min. To prevent the nonspecific binding of antibodies, a blocking solution called Blocking One Histo (06349-64, Nacala Tesque Inc., Kyoto, Japan) was applied. For the first immunostaining, the sections were incubated at 4 °C overnight with primary antibodies targeting γENaC (SPC-405D, Stress Marq Biosciences) and AQP2 (E-2, Santa Cruz Biotechnology), which was diluted to a concentration of 1:200 in PBS. After this incubation period, secondary antibodies (AlexaFluor 488 goat anti-rabbit IgG and AlexaFluor 555 donkey anti-mouse IgG, Thermo Fisher Scientific) were added at a dilution of 1:500 and incubated for 1 h. A fluorescence microscope BZ-X710 (Keyence Corporation, Osaka, Japan) was used to visualize the stained samples.

For the double immunostaining of plasminogen/plasmin and synaptopodin, frozen sections were fixed in acetone for 2 min and blocked with Blocking One Histo. Then, the sections were incubated overnight with anti-plasminogen (ab154560, Abcam) and anti-synaptopodin (sc-21537, Santa Cruz Biotechnology) primary antibodies, both diluted to a concentration of 1:500 in PBS. Subsequently, a combination of secondary antibodies, including AlexaFluor 488 goat anti-rabbit IgG and AlexaFluor 555 rabbit anti-goat IgG (1:500; Life Technologies, Carlsbad, CA, USA), along with DAPI solution (1:500; Dojindo, Kumamoto, Japan) for nuclear staining, were applied to the sections and incubated for 1 h at room temperature. The stained samples were visualized under a BZ-X710 microscope.

### 4.7. Terminal Deoxynucleotidyl Transferase dUTP Nick End Labeling (TUNEL) Assay

The frozen sections (3 μm thickness) were used for TUNEL staining, as previously described [15]. After overnight incubation with the anti-WT-1 (F-6) antibody (1:500, sc-7385, Santa Cruz Biotechnology) at 4 °C, the sections were further incubated with a secondary antibody (AlexaFluor 555 donkey anti-mouse IgG, 1:500) along with DAPI and the TUNEL reagent (in situ apoptosis detection kit; Takara, Shiga, Japan) according to the manufacturer’s instructions. This incubation step was carried out at 37 °C for 30 min. Subsequently, we counted the positive cells of WT-1 and TUNEL among 100 glomeruli per rat.

### 4.8. Quantitative Real-Time Polymerase Chain Reaction

The kidney tissue samples were preserved in RNA later (Sigma-Aldrich) at 4 °C overnight. The total RNA was eluted with the RNeasy mini kit (QIAGEN, Hilden, Germany). Reverse transcription of 1 μg of total RNA was performed using the PrimeScript RT reagent kit (Takara Bio Inc., Shiga, Japan). TaqMan probes specific for rat αENaC, βENaC, γENaC, EGF, EGFR, TNF-α, Fas, IL-1β, IL-6, and GAPDH were obtained from Applied Biosystems (Foster City, CA, USA), while the rat collagen-3 and fibronectin-1 probes were obtained from Sigma-Aldrich. The quantitative analysis was performed using the ∆Ct value (Ct value of the gene of interest minus Ct value of GAPDH). The relative gene expression analyses used the ΔΔCt method (∆Ct_sample_ minus ∆Ct_calibrator_).

### 4.9. Statistical Analyses

The results are reported as the average and standard deviation. We analyzed Figure 1A,B and Appendix A using two-way ANOVA, while we utilized one-way ANOVA for the other figures and Table 1. The post hoc analysis was performed using Tukey’s test. Statistical significance was set at *p* < 0.05. The statistical analysis was performed with GraphPad PRISM 9 software (GraphPad Software, La Jolla, CA, USA).

## Figures and Tables

**Figure 1 ijms-24-15743-f001:**
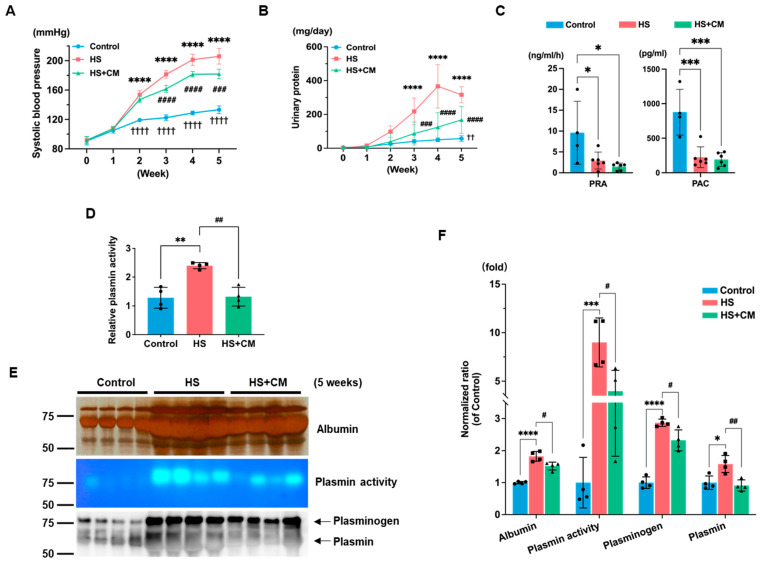
Effects of camostat mesilate (CM) on Dahl salt-sensitive (DS) rats with HS diet. (**A**) Weekly systolic blood pressure monitored with a tail-cuff method; n = 4, 6, and 6 for the control, high-salt (HS), and HS+CM groups. (**B**) The assessment of 24-h urinary protein excretion levels in every week; n = 4, 6, and 6. (**C**) PRA and PAC levels in DS rats of the respective groups at 5 weeks; n = 4, 6, and 6. (**D**) Urinary plasmin activities determined with a plasmin-specific substrate chromozym PL using urine sample at 5 weeks. The plasmin activity was shown as a fold change over the mean of the control group; n = 4 for each group. (**E**) Silver-stained urinary albumin and Western blot analysis of plasminogen and plasmin. Urinary plasmin activity was assessed using double-layer fluorescent zymography; n = 4 for each group. (**F**) Western blot results analyzed with Image J. The results are reported as the average and standard deviation. We analyzed (**A**,**B**) using two-way ANOVA, while we utilized one-way ANOVA for the other figures, followed by Tukey’s test. *: *p* < 0.05, **: *p* < 0.01, ***: *p* < 0.001, ****: *p* < 0.0001 vs. control group. #: *p* < 0.05, ##: *p* < 0.01, ###: *p* < 0.001, ####: *p* < 0.0001 vs. HS group. ††: *p* < 0.01, ††††: *p* < 0.0001, control group vs. HS+CM group.

**Figure 2 ijms-24-15743-f002:**
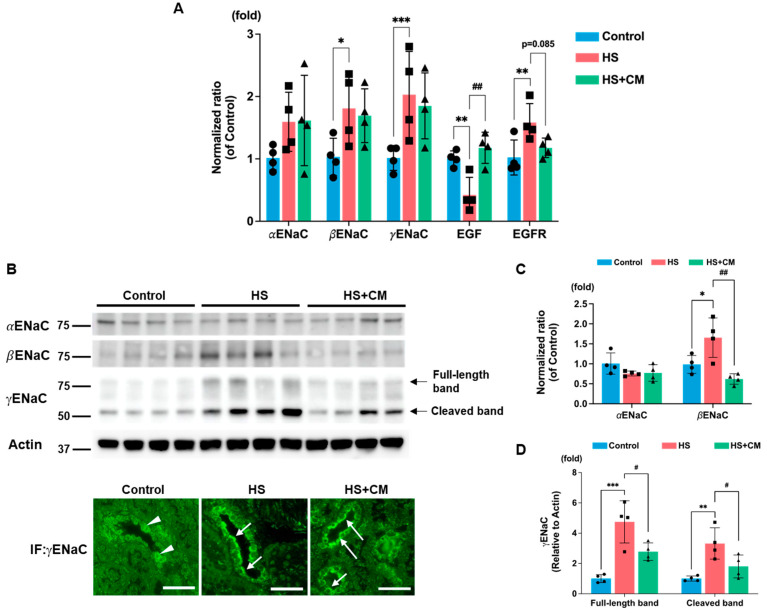
Effects of camostat mesilate (CM) on the renal expression of epithelial sodium channels (ENaC), epidermal growth factor (EGF), and EGF receptor (EGFR). (**A**) renal mRNA expression levels of three subunits of ENaC, EGF, and EGFR; n = 4 for each group. (**B**) Immunoblotting of the three ENaC subunits in the renal membrane fraction, and immunofluorescence staining showing the distribution of γENaC in distal renal tubules. The arrowhead and arrow indicate cytoplasmic and apical membrane localization of γENaC, respectively (microscopic magnification at 400×). White scale bar represents 50 μm. (**C**,**D**) Scatter plots summarizing the quantitative analysis of αENaC, βENaC, and full-length band and cleaved band of γENaC; n = 4 for three groups. The results are reported as the average and standard deviation. We analyzed results using one-way ANOVA, followed by Tukey’s test. *: *p* < 0.05, **: *p* < 0.01, ***: *p* < 0.001 vs. control group. #: *p* < 0.05, ##: *p* < 0.01 vs. high-salt (HS) group.

**Figure 3 ijms-24-15743-f003:**
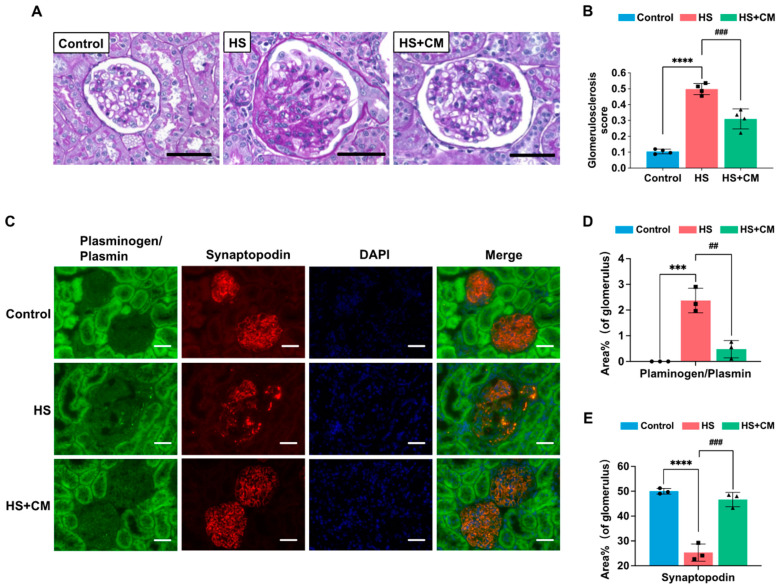
Effects of camostat mesilate (CM) on glomerular sclerosis and co-immunostaining of plasminogen/plasmin and the podocyte marker synaptopodin in glomeruli. (**A**) Representative images of PAS staining of the kidney (microscopic magnification at 400×). Black scale bar represents 50 μm. (**B**) Determination of glomerulosclerosis scores; n = 3 for each group. (**C**) Co-immunostaining of plasminogen/plasmin, synaptopodin, and nuclear marker DAPI (microscopic magnification at 400×). White scale bar represents 50 μm. Quantification of plasminogen/plasmin (**D**) and synaptopodin (**E**) areas in 30 glomeruli per rat measured with ImageJ; n = 3 for each group. The results are reported as the average and standard deviation. We analyzed results using one-way ANOVA, followed by Tukey’s test. ***: *p* < 0.001, ****: *p* < 0.0001 vs. control group. ##: *p* < 0.01, ###: *p* < 0.001 vs. high-salt (HS) group.

**Figure 4 ijms-24-15743-f004:**
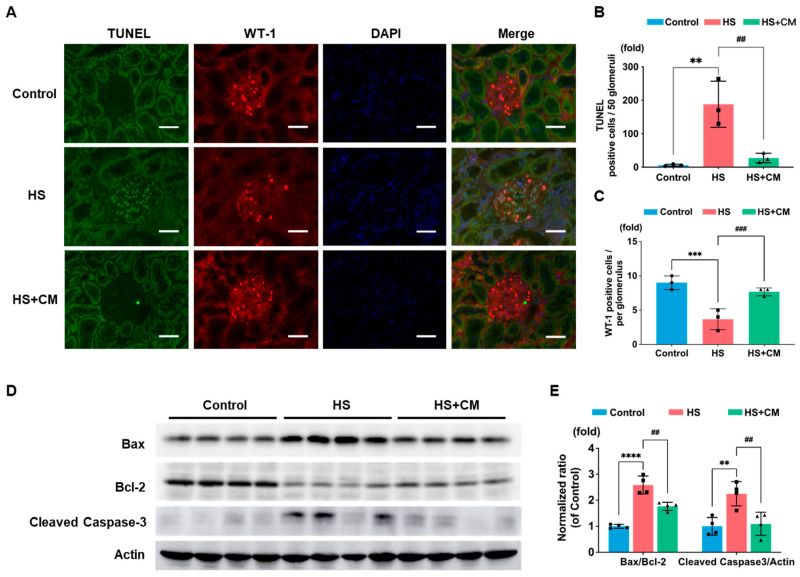
Effects of camostat mesilate (CM) on podocyte apoptosis. (**A**) Co-immunostaining for TUNEL (green), podocyte marker WT-1 (red), and DAPI (blue) (microscopic magnification at 400×). White scale bar represents 50 μm. Quantitative analysis of the number of (**B**) TUNEL-positive and (**C**) WT-1-positive cells; n = 3 for each group. (**D**) Immunoblotting of Bax, Bcl-2, and cleaved caspase-3 expressions in the kidney. (**E**) Densitometry analysis of Bax/Bcl-2 ratio and cleaved caspase-3/actin levels; n = 4 for each group. The results are reported as the average and standard deviation. We analyzed results using one-way ANOVA, followed by Tukey’s test. **: *p* < 0.01, ***: *p* < 0.001, **** *p* < 0.0001 vs. control group. ##: *p* < 0.01, ###: *p* < 0.001 vs. high-salt (HS) group.

**Figure 5 ijms-24-15743-f005:**
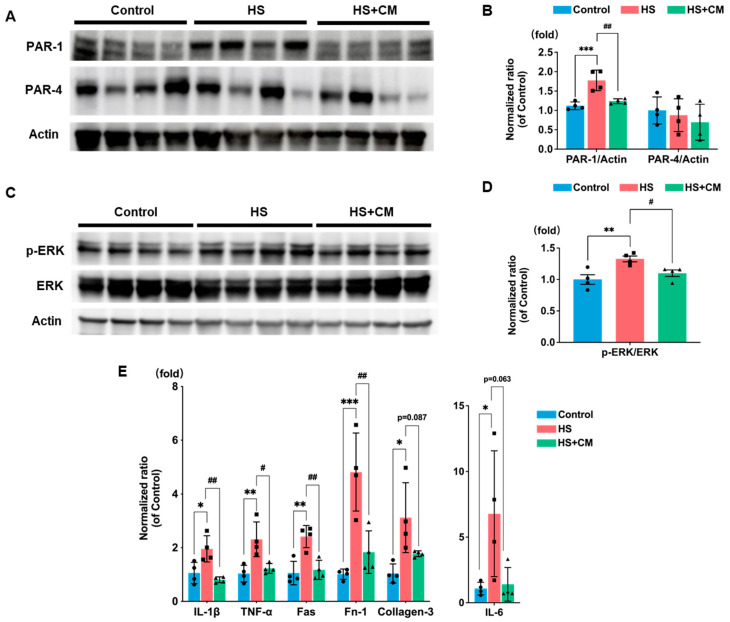
Effects of camostat mesilate (CM) on protease-activated receptor (PAR) expression, downstream signaling pathway, and mRNA expressions of renal injury markers in the kidney. (**A**) Western blot analysis of PAR-1 and PAR-4 expressions; n = 4 for each group (control, high-salt (HS), and HS+CM). (**B**) Densitometry analysis of PAR-1 and PAR-4 levels that were corrected with actin. Fold increases over the control group are presented. (**C**) Immunoblotting of extracellular signal-regulated kinase (ERK) and its phosphorylation (p-ERK) levels in the kidney. (**D**) Densitometry analysis of p-ERK/ERK levels; n = 4 for each group. (**E**) mRNA expression levels of IL-1β, IL-6, TNF-α, Fas, Fn (fibronectin)-1, and collagen-3 evaluated via real-time PCR. The gene expression levels were normalized to GAPDH in each rat, and presented as fold-change relative to the control group; n = 4 for each group. The results are reported as the average and standard deviation. We analyzed results using one-way ANOVA, followed by Tukey’s test. *: *p* < 0.05, **: *p* < 0.01, ***: *p* < 0.001 vs. control group. #: *p* < 0.05, ##: *p* < 0.01 vs. high-salt (HS) group.

**Figure 6 ijms-24-15743-f006:**
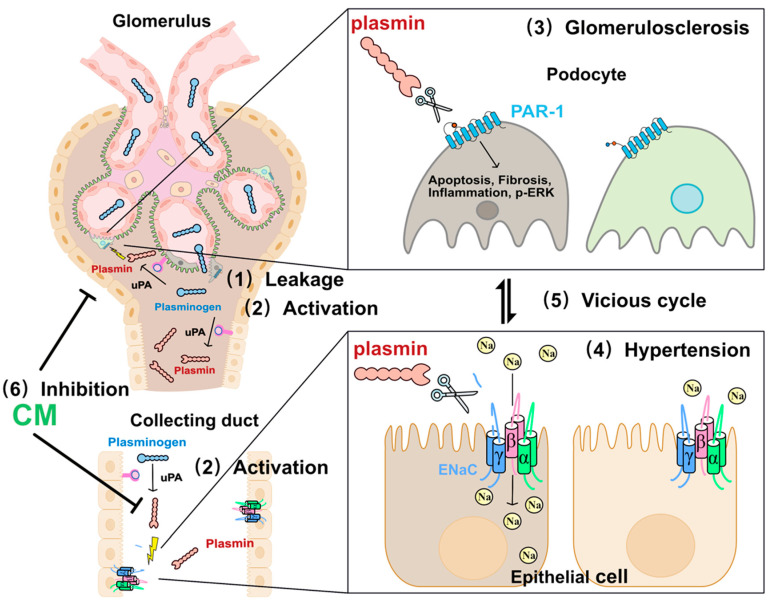
Summary of the mechanisms underlying plasmin-induced ENaC activation and podocyte injury, and the effects of CM on them in salt-sensitive hypertension. Plasminogen leakage from damaged glomerular filtration barrier is converted to an active serine protease plasmin, and plasmin activates ENaC and provokes podocyte injury, which leads to salt-sensitive hypertension and glomerular sclerosis. The synthetic serine protease inhibitor, camostat mesilate, alleviates these changes. uPA: urokinase-type plasminogen activator, PAR-1: protease-activated receptor-1.

**Table 1 ijms-24-15743-t001:** Relative organ weight and blood test. The heart and kidney were excised under anesthetic conditions, and then the weight (Wt) of each organ was corrected for body weight (BW). TP; total protein, Cr; creatinine. The results are reported as the average and standard deviation.; n = 4, 6, and 6 for the control, high-salt diet (HS), and HS + camostat mesilate (CM) groups. *****: *p* < 0.05, *******
*p* < 0.001, ********
*p* < 0.0001 vs. control group. **#**: *p* < 0.05, **##**: *p* < 0.01 vs. HS group. **†**
*p* = 0.058 vs. control group. **‡**
*p* = 0.059 vs. HS group.

Group	Control	HS	HS+CM
Relative organ weight
Heart Wt/BW (g)	3.7 ± 0.2	5.3 ± 0.6 *******	4.6 ± 0.3 ***^,‡^**
Kidney Wt/BW (g)	8.1 ± 0.5	11.8 ± 0.7 ********	10.9 ± 0.8 ******^,#^**
Blood test
TP (mg/dL)	6.0 ± 0.7	6.0 ± 0.5	5.6 ± 0.7
Cr (mg/dL)	0.27 ± 0.04	0.40 ± 0.08 *****	0.30 ± 0.03 **^#^**
Na (mEq/L)	143.0 ± 1.4	144.8 ± 1.5 **^†^**	140.8 ± 1.9 **^##^**
Cl (mEq/L)	99.0 ± 1.4	100.7 ± 2.4	97.3 ± 2.0 **^#^**
K (mEq/L)	4.4 ± 0.2	4.3 ± 0.4	4.2 ± 0.4

## Data Availability

Data are available upon request.

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
