# Peer review of "A Serine Protease Inhibitor, Camostat Mesilate, Suppresses Urinary Plasmin Activity and Alleviates Hypertension and Podocyte Injury in Dahl Salt-Sensitive Rats"

_ijms, 2023, doi:10.3390/ijms242115743_

Round 1
Reviewer 1 Report
Comments and Suggestions for Authors
Iwata et al. report that camostat mesylate suppresses urinary plasmin activity and alleviates hypertension and podocyte injury in hypertensive rat models. This experimental work has novel approaches to regulating urinary plasmin to target podocyte and tubular epithelial cells.
I have minor comments on this work.
1. In Figure 2B, the immunofluorescence stain for γENaC should have a double stain for a marker of the distal tubules.
2. In Figure 4, the authors show the TUNEL-positive cells in glomeruli with the double stain of WT1. Some TUNEL-positive cells are merged with WT1, while others are not. So, What kind of cells have TUNEL-positivity? Authors should present with other mesangial cell markers with TUNEL assay to address this question.
3. In Figure 4D, The authors use a whole kidney sample for Western blot analysis. Why do the authors use whole kidneys for Bax, Bcl2, and cleaved caspase expression? How about using isolated glomeruli?
Comments on the Quality of English LanguageMinor English editing before publication
Author Response
We wish to express our appreciation to the Reviewer 1 for his or her insightful comments, which have helped us significantly improve the paper. Please check our point-by-point responses to each of your comments.
- In Figure 2B, the immunofluorescence stain for γENaC should have a double stain for a marker of the distal tubules.
Response:
We appreciate your comment and the suggestion to provide a double stain of γENaC and a marker of the distal tubules. In response to your suggestion, we conducted double staining for γENaC and AQP2, a representative marker of the collecting duct. It was confirmed that γENaC and AQP2 co-localize in the same cells (Figure S2).
We hope that this addition addresses your concern and enhances the comprehensiveness of our study. We thank the reviewer 1 for pointing out this issue.
- In Figure 4, the authors show the TUNEL-positive cells in glomeruli with the double stain of WT1. Some TUNEL-positive cells are merged with WT1, while others are not. So, What kind of cells have TUNEL-positivity? Authors should present with other mesangial cell markers with TUNEL assay to address this question.
Response:
Thank you for your feedback regarding this issue. Because WT-1 is a marker of healthy podocytes and it is reduced in injured podocytes, the most of TUNEL-positive podocytes might have already lost WT-1 staining. Therefore, it is difficult to observe the double staining of both markers in same single podocyte. In Figure 4, we observed the merge of both markers in some podocytes, suggesting that some of TUNEL-positive cells were podocytes. In this experiment, we were unable to use a specific marker for mesangial cells, and therefore, we could not definitively confirm whether the observed cells were mesangial cells or not.
In response to your suggestion, we inserted the following sentences in the manuscript, “Because WT-1 is a marker of healthy podocytes and it is reduced in injured podocytes, the most of TUNEL-positive podocytes might have already lost WT-1 staining. Therefore, it is difficult to observe the double staining of both markers in same single podocyte. (Page 7, line 173-176)”
We thank the reviewer 1 for pointing out this issue.
- In Figure 4D, the authors use a whole kidney sample for Western blot analysis. Why do the authors use whole kidneys for Bax, Bcl2, and cleaved caspase expression? How about using isolated glomeruli?
Response:
Thank you for your feedback regarding this issue. As the Reviewer 1 pointed out, we also believe that it would be more accurate to isolate glomeruli; however, due to the need to collect kidney tissues using multiple methods simultaneously, we did not have the capacity to isolate glomeruli. In immunostaining, most of the apoptotic cells were located within the glomeruli, not renal tubules, and we believe that the observed changes in apoptosis-related proteins in the whole kidney reflect the changes occurring in the glomeruli.
We thank the reviewer 1 for pointing out this issue.
Reviewer 2 Report
Comments and Suggestions for Authors
In the manuscript by Iwata et al., A serine protease inhibitor, Camostat Mesilate, suppresses urinary plasmin activity and alleviates hypertension and podocyte injury in Dahl salt-sensitive rats, authors have evaluated extensively evaluated effect of Camostat Mesilate on blood pressure and plasmin activity in anima model of hypertension. Although it is well designed and executed study, I have following concern:
1. In figure 1 A-B (N, 4,6,6) while Figure 1C-F, Figure 2 (N=4,4,4), what is rational for same.
2. Fig 2B: Please provide lower magnification in immunofluorescence staining to cover more area.
3. Discussion: It is more repetition of results.
4. The dose and route of DM is missing.
5. Figure 3: N=3 is not sufficient for concluding effect. Please provide lower magnification in immunofluorescence staining to cover more area.
6. There is lack of clarity for how kidneys were used for various experiments, WB, Immunofluorescence as well as histology.
7. Pleas provide unedited blot for WB
8. ARRIVE guideline: Nowadays it is recommended to follow ARRIVE guidelines for animal experimentation.
9. Statistical analysis: In each figure legend, please write details Statistical test. In Figure 1AB and S1 appropriate test was not used.
Author Response
We wish to express our appreciation to the Reviewer 2 for his or her insightful comments, which have helped us significantly improve the paper. Please check our point-by-point responses to each of your comments.
- In figure 1 A-B (N, 4,6,6) while Figure 1C-F, Figure 2 (N=4,4,4), what is rational for same.
Response:
Thank you for your feedback regarding this issue. With regard to Figure 1C, we replaced N=4,4,4 with N=4,6,6, and we have observed similar results. However, due to constraints on the number of lanes on the SDS-PAGE gel, Figure E has been presented with representative images for n=4, 4, 4. That is consistent with the other figures for Western blotting.
We thank the reviewer 2 for pointing out this issue.
- Fig 2B: Please provide lower magnification in immunofluorescence staining to cover more area.
Response:
Thank you for your feedback regarding Figure 2B and the suggestion to provide lower magnification in immunofluorescence staining to cover a broader area. In response to your comment, we have included lower magnification images (100×) in the revised supplementary Figure 2 to provide a broader view of the area. In addition, we conducted double staining for γENaC and AQP2, a representative marker of the collecting duct. The double staining confirmed the colocalization of γENaC and AQP2 in the collecting duct.
We hope that this addition addresses your concern and enhances the comprehensiveness of our study. We thank the reviewer 2 for pointing out this issue.
- Discussion: It is more repetition of results.
Response:
To address your comments, we have made additions to the discussion section, particularly about the limitation of our study (Page 10, line 285- Page 11, line 323). Please review the revised manuscript, and we hope that these changes align with your requests.
We thank the reviewer 2 for pointing out this issue.
- The dose and route of DM is missing.
Response:
Thank you for your feedback regarding the concentration of CM used in our study. In response to your comment, we have included a statement in the main text, specifying that “CM was administered orally by mixing it into the diet at a concentration of 0.1%” in Materials and methods (Page 12, line 348-349). This addition clarifies the method of administration, ensuring transparency in our research.
We thank the reviewer 2 for pointing out this issue.
- Figure 3: N=3 is not sufficient for concluding effect. Please provide lower magnification in immunofluorescence staining to cover more area.
Response:
We appreciate your comment and the suggestion to provide lower magnification in immunofluorescence staining to cover a broader area in Figure 3, as well as your concern regarding the sample size (N=3). In response to your feedback, we have incorporated lower magnification images in the revised Supplementary Figure 3 to ensure a more comprehensive view of the area of interest. These new images provide a broader perspective. We would like to emphasize that the observed trends are consistent with Figure 3. In the HS group, plasminogen/plasmin adheres to the glomerulus and co-localizes with synaptopodin, which appears reduced due to glomerular injury.
We thank the reviewer 2 for pointing out this issue.
- There is lack of clarity for how kidneys were used for various experiments, WB, Immunofluorescence as well as histology.
Response:
Thank you for your feedback and the request for clarification regarding the processing of samples for each analysis. We appreciate your input, and in response to your comment, we have made the necessary additions to the main text. We have now included detailed information on how sample for each analysis was processed in the manuscript, enhancing the clarity of the Materials and methods section. In Western blotting, we described how membrane proteins, whole proteins, and phosphorylated proteins were extracted and specified which molecules were observed for each protein. Additionally, as you mentioned, information regarding sample processing for techniques such as histology, immunostaining and PCR was already present in the main text.
If there are any further areas that require attention or clarification, please let us know. We are committed to addressing any additional concerns you may have. We are grateful for your careful review.
- Pleas provide unedited blot for WB
Response:
We have already submitted the unedited Western Blot as per Assistant Editor’s similar request. Please kindly review the submitted PDF.
We thank the reviewer 2 for pointing out this issue.
- ARRIVE guideline: Nowadays it is recommended to follow ARRIVE guidelines for animal experimentation.
Response:
Thank you for your valuable feedback and recommendation regarding the ARRIVE guidelines for animal experimentation. We appreciate your concern for the quality and transparency of our research.
In response to your comment, we want to assure you that we have closely followed the ARRIVE guidelines in the planning and execution of our animal experiments. We have taken care to report all relevant details in our manuscript to enhance the transparency of our methods. Therefore, we revised the manuscript as below: “All animal experiments were conducted following the approved guidelines at Kumamoto University (No. A2021-096) and ARRIVE (Animal Research: Reporting of In Vivo Experiments) guidelines. (Page 12, line 340-342)”
If there are specific aspects of the ARRIVE guidelines that you believe need further attention or clarification in our manuscript, please let us know. We are committed to addressing any concerns and ensuring that our research meets the highest ethical and scientific standards.
We thank the reviewer 2 for pointing out this issue.
- Statistical analysis: In each figure legend, please write details Statistical test. In Figure 1AB and S1 appropriate test was not used.
Response:
Thank you for your feedback regarding the statistical analysis in this manuscript. In response to your comment, we analyzed Figure 1A, 1B and Supplementary Figure 1 using two-way ANOVA, while we utilized one-way ANOVA for the other figures and Table. Post hoc analysis was per-formed using the Tukey test. We included descriptions of the statistical analysis in the 'Materials and methods' section and in the respective figure legends.
We thank the reviewer 2 for pointing out this issue.
Round 2
Reviewer 2 Report
Comments and Suggestions for Authors
I have no further comments, the authors have addressed all comments.